# Endoplasmic Stress Affects the Coinfection of Leishmania Amazonensis and the Phlebovirus (Bunyaviridae) Icoaraci

**DOI:** 10.3390/v14091948

**Published:** 2022-09-02

**Authors:** José V. dos Santos, Patricia F. Freixo, Áislan de C. Vivarini, Jorge M. Medina, Lucio A. Caldas, Marcia Attias, Karina L. Dias Teixeira, Teresa Cristina C. Silva, Ulisses G. Lopes

**Affiliations:** 1Laboratory of Molecular Parasitology, Universidade Federal do Rio de Janeiro, Rio de Janeiro 21941-902, RJ, Brazil; 2Laboratory of Ultrastructure Hertha Meyer Cellular, Universidade Federal do Rio de Janeiro, Rio de Janeiro 21941-902, RJ, Brazil; 3Department of Cell Biology, University of Virginia, Pinn Hall, Charlottesville, VA 22908, USA

**Keywords:** *Phlebovirus*, *Leishmania*, ER stress

## Abstract

Viral coinfections can modulate the severity of parasitic diseases, such as human cutaneous leishmaniasis. Leishmania parasites infect thousands of people worldwide and cause from single cutaneous self-healing lesions to massive mucosal destructive lesions. The transmission to vertebrates requires the bite of Phlebotomine sandflies, which can also transmit Phlebovirus. We have demonstrated that Leishmania infection requires and triggers the Endoplasmic stress (ER stress) response in infected macrophages. In the present paper, we tested the hypothesis that ER stress is increased and required for the aggravation of *Leishmania* infection due to coinfection with *Phlebovirus*. We demonstrated that *Phlebovirus* Icoaraci induces the ER stress program in macrophages mediated by the branches IRE/XBP1 and PERK/ATF4. The coinfection with *L. amazonensis* potentiates and sustains the ER stress, and the inhibition of IRE1α or PERK results in poor viral replication and decreased parasite load in macrophages. Importantly, we observed an increase in viral replication during the coinfection with *Leishmania*. Our results demonstrated the role of ER stress branches IRE1/XBP1 and PERK/ATF4 in the synergic effect on the Leishmania increased load during *Phlebovirus* coinfection and suggests that *Leishmania* infection can also increase the replication of *Phlebovirus* in macrophages.

## 1. Introduction

Cells may face stress conditions due to various sources of extracellular or intracellular factors. The cellular stress response is an adaptive process to reestablish cell homeostasis. The Endoplasmic Reticulum (ER) houses different processes, such as protein folding, lipid synthesis, and the biogenesis of autophagosomes and peroxisomes [1,2]. Whenever a stress condition interferes with protein synthesis or folding, ER initiates transcription and translational response, which involves lipid synthesis, the upregulation of chaperone expression, and the coordination of the retrograde transport and degradation of unfolded proteins (ERAD), [3].

The ER stress response comprises three branches: the transcription factor 6 (ATF6), inositol-requiring kinase 1 (IRE-1), and the protein kinase R (PKR)-like endoplasmic reticulum kinase (PERK). These arms of ER stress initiate with the engagement of ER sensors and culminate with the activation of a set of transcription factors [4]. The sensor IRE-1 activates the X-box binding protein 1 (XBP-1) due to the cytoplasmic splicing of the X-box binding protein 1 (XBP-1) RNA, leading to the translation of the mature XBP-1 RNA [5,6]. The protein kinase R (PKR)-like (PERK) phosphorylates the initiation translational factor eIF2α, which restricts the overall protein translation and upregulates the expression of activating transcription factor 4 (ATF4) [7,8]. The reactivated translation of ATF-4-dependent genes may generate significant amounts of reactive oxygen species (ROS), leading to cell death. To reestablish cell homeostasis, PERK/ATF4 controls the expression of the nuclear factor erythroid 2-related factor 2 (Nrf2) that regulates the expression of anti-oxidative responsive genes [8,9]. The ER stress plays a significant role in inflammatory processes [10], tumor progression [11], in secretory cells, such as B-lymphocytes [12], and in infections [13].

Leishmaniasis is an emerging parasitic disease caused by several species of *Leishmania* and affects ~12 million people worldwide, with approximately 2 million new infections per year [14]. Depending on the species and the host immune response, diverse clinical manifestations can be observed ranging from cutaneous to visceral form of the infection. Leishmania parasites reach the vertebrate hosts through the regurgitation and inoculation of metacyclic promastigotes by Phlebotomine sandfly vectors. Inside the vertebrate hosts, promastigotes replicate as amastigotes in macrophages, subverting the immune response [15,16].

*Leishmania amazonensis* is endemic in Brazil and is the causative agent of local cutaneous leishmaniasis (LCL) and a more severe form, the anergic diffuse cutaneous leishmaniasis (ADCL). The immunological response is associated to a T-cell hyposensitivity pole and patients present a nonreactive profile characterized by negative delayed type hypersensitivity (DTH), [17]

We demonstrated that *Leishmania amazonensis* promotes the ER stress response in infected macrophages, and immunohistological analysis of human clinical samples showed the expression of XBP-1 and ATF4 [18,19]. Another group reported similar results in *L. infantum* infected macrophages [20]. The Amazon Region presents different *Leishmania* species with distinct sylvatic cycles, covering a diversity of sandflies vectors and natural reservoirs. *L amazonensis* distribution was initially described in some regions of the North of Brazil. However, autochthonous transmission may take place in other regions, including periurban areas [21].

The Brazilian Amazon region is probably the largest arboviruses reserve in the world [22], which may favor coinfections due to the convergence of hosts and vectors shared by Leishmania and arbovirus transmitted by sandflies.

*Leishmania* vectors can also transmit arbovirus belonging to the Family Bunyaviridae, and they are known as *Phlebovirus* [23]. Phlebovirus are now in the Phenuiviridae family, the term Bunyaviridae has been substituted with Phenuiviridae [24,25], causing distinct symptoms [26,27]. Some studies showed overlapped phleboviruses and *Leishmania* transmission areas, and sand fly coinfections were reported [28,29]. Our previous studies utilizing in vivo and in vitro models demonstrated that an Amazonian Phlebovírus (Icoaraci, ICOV) which shares the same vector and wild reservoir of *L. amazonensis*, potentiated the parasite infection [30].

The present paper reports that ICOV induces and enhances a sustained ER stress response in coinfection with *L. amazonensis*, which is partially required for the aggravation of the parasitic infection. Importantly, ICOV replication is also favored in the coinfection of macrophages with *Leishmania*, thus suggesting a mutual benefit of the pathogens inside vertebrate cells.

## 2. Materials and Methods

### 2.1. Peritoneal Macrophages

Thioglycolate-elicited peritoneal macrophages from eight-week-old wild-type (WT)

C57BL/6 mice were elicited with 2 mL of 3% sodium thioglycolate Brewer solution for 4 days and obtained by injecting PBS into the peritoneal cavity. The cell suspension was washed in PBS once and then resuspended in serum-free DMEM. Cells were plated on glass coverslips at 2 × 105/well in 24-well polystyrene plates or 4 × 106/well in 6-well polystyrene plates and incubated for 1 h at 37 °C in a 5% CO_2_ atmosphere. Nonadherent cells were washed out with PBS, and the adherent cell population was incubated overnight in DMEM with high glucose and supplemented with 10% heat-inactivated FBS, 100 U/mL penicillin, and 100 g/mL streptomycin for subsequent treatment or Leishmania/virus infection assays.

### 2.2. Cell Culture

BHK21 cell lines (Baby hamster kidney cell -ATCC: CCL-10) were kindly provided by Dr. Clarissa Damaso (Instituto de Biofísica Carlos Chagas Filho—UFRJ). Cells were grown in 25 cm^2^ cell culture flasks at 37 °C in Dulbecco’s modified Eagle’s medium high glucose (DMEM—Gibco) supplemented with 10% of heat-inactivated fetal bovine serum and 1% penicillin/streptomycin.

### 2.3. Phlebovirus: Production, Quantification, and Infection

The *Phlebovirus* Icoaraci (BeAN 24262—ICOV) used in this work were obtained through Dr. Pedro Vasconcelos (Instituto Evandro Chagas/SVS/MS). Viruses from lyophilized mouse brain macerates were passaged five times in BHK-21 cells to recover infectivity. For viral propagation, BHK-21 cells were seeded in 100-mm plates and the virus inoculum was adsorbed for 2 h at room temperature with a MOI (multiplicity of infection) of 0.01. The inoculum was removed and the cells were maintained in DMEM with high glucose and supplemented with 10% heat-inactivated FBS at 37 °C for 3 days. For viral concentration, cell culture supernatants containing viral particles were collected at 18,000× *g* for 1 h at 4 °C. The pellet was reconstituted in PBS (Gibco) and stored at −80 °C. Viral titers were determined by plaque assay. BHK-21 cells were seeded in 6-well polystyrene plates (initial inoculum of 1.5 × 10^6^ cells/well). After 24 h, the cells were infected with a 10-fold of serial dilution of ICOV stock and incubated for 2 h at room temperature. Subsequently, the inoculum was removed and the monolayer covered with semisolid medium composed of DMEM with high glucose supplemented with 10% FBS and 1% methylcellulose (Sigma-Aldrich, Saint Louis, MO, USA). Viral plaques were observed 3 days after infection when cells were fixed and stained with a 0.1% crystal violet solution containing 10% formaldehyde. Viral plaques were counted and titers were expressed as PFU/mL. For macrophage infections, adsorption was performed for 1 h at 37 °C using a MOI of 1/1. Infection times varied according to the objective of the experiment, [30,31].

### 2.4. Parasites, Culture Conditions, and Infection

*L. (L) amazonensis* (WHOM/R/75/Josefa) was maintained in vitro in Schneider’s Insect Medium (Sigma) supplemented with 10% of fetal bovine serum. The infection was carried out with stationary phase promastigotes at a parasite-to-cell ratio of 5:1 in triplicate. For the survival assays, peritoneal macrophages were infected for 48 h at 35 °C, followed by washing, fixation with methanol, and Giemsa staining. Infected macrophages were counted by light microscopy to assess the infection index as follows: 100 Giemsa-stained cells were inspected and the percentage of infected macrophages was multiplied by the average number of amastigotes per macrophage. Macrophages were treated with 40 nM of the PERK inhibitor (cat GSK2606414—MERCK.) or 100 nM IRE1α inhibitor (cat 4μ8c 14003-96-4 MERK).

### 2.5. Conditions for ICOV/Leishmania Coinfection

Macrophages were plated on glass coverslips in 24-well plates (2 × 10^5^ per well) or 6-well plates (4 × 10^6^ per well). After 1 h of adherence at 37 °C, cells were washed with PBS and cultured with DMEM containing 10% FBS (Gibco), 2% (*v/v*) Penicillin (10 μ/mL), and Streptomycin (10 μg/mL). Macrophages rested for two days at 37°C in an atmosphere of 5% CO_2_. For the peritoneal macrophages, ICOV adsorption was performed for 1 h at 37 °C using MOI (Multiplicity of Infection) of 1.

After ICOV adsorption, the cultures were washed with PBS, and the macrophages were infected with promastigotes at the stationary phase with a ratio of 5:1.

### 2.6. En Bloc Processing for Transmission Electron Microscopy (TEM)

For TEM analysis, mock and infected macrophages in 25 cm^2^ plastic culture flasks were fixed in 2.5% glutaraldehyde in 0.1 M cacodylate buffer (pH 7.2) and post-fixed for 1 h in 1% OsO_4_ 0.8% potassium ferrocyanide in the same buffer. Then, samples were incubated with 2.5% uranyl acetate in water for 2 h, washed three times in 0.1 M cacodylate buffer (pH 7.2), dehydrated in ethanol, and embedded in Polybed resin (Polysciences). Spatial orientation of the samples was maintained for ultrathin sections, which were stained with 5% uranyl acetate (40 min) and 4% lead citrate (5 min) before observation using Hitachi HT7800 transmission electron microscope equipped with an AMT CMOS 8K camera.

### 2.7. Quantitative RT-PCR

Total RNA of peritoneal macrophages was extracted with RNeasyR Plus (Quiagen) and 1μg total RNA was reverse-transcribed into the first-strand cDNA with ImProm (Promega) and oligo(dT) 12–18 primer, according to the manufacturer’s instructions. The following pairs of primers were used to determinate the mRNA levels of uXBP1: Forward 5′CCGCAGCACTCAGACTATG 3′-Reverse 5′GGGTCCAACTTGTCC AGAAT3′; sXBP1: Forward 5′ CTGAGTCCGCAGCAGGT3′-Reverse 5′AAACATGACAGGGTC CAACTT3′; ATF4 Forward 5′AGCACTCAGACTACGTGCACCTCT3′-Reverse 5′GAAGAGTCAATACCGCCAGAATCC3′; N protein gene of ICOV: Forward 5′AGGTGAGGCTGTAAATCTTG3′-Reverse 5′TCACATCATCCTTCCAAGTG3′; Chpt1: Forward: 5′GGAGGAGGCACCATACTGGACATA3′-Reverse: 5′CGCCGATGGCCATAAATACTGTGG3′; Cept1: Forward: 5′GAGTACCCTCATGGATTGCCCCA3′-Reverse: 5′CCCACAGAGGTGCCTGCT 3′; NRF2: Forward 5′TGCACCACCAACTGCTTAGC3′-Reverse 5′GGCATGGCATGTGGTCATGAG3′; GAPDH: Forward 5′GTGTCCGTCGTGGATCTG3′-Reverse 5‘TGCTTCACCACCTTC TTGA3′ (used for normalization). Real time quantitative PCR (qPCR) was carried out via the Applied Biosystems StepONE system using Power SYBR Green PCR Master Mix (Applied Biosystems). All qPCR experiments were performed at least three times and the expression ratios were determined by the relative gene expression ΔΔCt method using StepOne 2.0 software (Applied Biosystems, Waltham, MA, USA).

### 2.8. Cell Viability Assay

Peritoneal macrophages were seeded in a 24-well plate. After 24 h, cells were treated with PERK inhibitor (40 nM GSK) or IRE1α inhibitor (100 nM 4µ8c). Cultures were maintained at 37 °C in a 5% CO_2_ atmosphere for 48 h. The culture medium was removed and 100 μL of MTT (3-(4,5-dimethylthiazol-2-yl)-2,5-diphenyltetrazolium bromide) (1 mg/mL) was added. After 1 h of incubation at 37 °C, MTT was removed and 250 µL of isopropyl alcohol was added. Samples were homogenized and optical density was measured at 570 nm.

### 2.9. Statistical Analysis

Data were analyzed by two-way ANOVA for independent samples followed by a post hoc Bufferoni compare all columns (with no designated control group), using GraphPad Prism 5 software (San Diego, CA, USA). Data were presented as the mean values ± standard error of the mean (SEM) of three independent experiments. Comparisons between means were considered to be statistically significant when *p* < 0.05.

## 3. Results

### 3.1. Ultrastructure Aspects of ICOV in Murine Macrophages

We pursued TEM experiments in macrophages infected by ICOV to get insights into the interaction with macrophages, which are primary cells infected by *Leishmania*.

Macrophages infected with ICOV (MOI = 1) were processed at 24 h post-infection for visualization by transmission electron microscopy (TEM). At this time, viral particles were observed attached to the cell surface (Figure 1A,B). However, ICOV seemed to be also present in the lumen of large vacuoles (Figure 1C,D). In another macrophage, the interaction between the virus and the inner face of the vacuolar membrane displayed a belt-like structure, suggesting the virus pinch-off that allows the virus to occupy the lumen of the vacuole.

ICOV was also observed in canyons adjacent to cell membrane protrusions (Figure 1E,F), a site of extensive actin activity. Discontinuation of cell plasma membrane indicates the viral egress. In the same image, the en bloc contrasting that makes the viral nucleic acid more electrondense for visualization by TEM revealed a virus-like-particle, tightly adhered to the cell surface.

### 3.2. ICOV Enhances Leishmania Growth through the ER Stress Response

Viral infections induce and may require ER the stress response to replicate inside infected cells [32,33]. We decided to test whether the inhibition of IRE1α or PERK would affect the exacerbated parasitic growth due to ICOV coinfection. Figure 2A,B confirms our previous findings on increasing the *L. amazonensis* load and the number of infected macrophages upon viral coinfection, which is comparable with the induction of ER stress by thapsigargin. Then, we tested the role of IRE1α in the context of coinfection. Figure 2C,D shows the reduction of the parasite load by the inhibitor, even in the presence of ICOV.

We decided to address whether PERK/ATF4 would play a role in the coinfection. Figure 1E,F shows the reduction of the parasite load and the percentual of infected macrophages 48 h post coinfection due to the inhibition of PERK. Macrophage and Leishmania viability were not affected during treatment with IRE1α and PERK inhibitors (Appendix A).

We concluded that two branches of the ER response, IRE1α and PERK/ATF4, are necessary to enhance parasitic growth in the ICOV coinfection model.

### 3.3. The ER Stress Branch IRE1α/XBP1 Is Required for ICOV Replication, and It Is Augmented and Sustained during Coinfection

IRE1a links the ER to the nucleus by modulating the XBP-1 activity and the expression of several genes involved in the Unfolded Protein Response (UPR). IRE1 alpha protein possesses intrinsic kinase and endoribonuclease activity; the latter is responsible for the splicing of unprocessed uXBP1 to sXBP1 mRNA, which will be translated to the active XBP1 transcription factor [34].

We sought to investigate if ICOV induces IRE1a/XBP1 signaling during the infection. Figure 3A shows the expression of sXBP1 in ICOV infected macrophages. Then, we tested whether the coinfection by ICOV/L.amazonensis would affect the levels of sXBP1mRNA. Interestingly, the coinfection leads to increased levels of sXBP1 mRNA within 6 h of infection (Figure 3B), and it sustains at the later time point of 24 h, Appendix A.

We decided to test whether the coinfection would benefit viral replication. Figure 3C shows the increase in the number of viral particles measured by plaque assays in the coinfection condition. Figure 3D shows the rise in the mRNA for the structural viral protein N during coinfection, thus supporting the viral plaque results. Notably, the chemical inhibition of IRE1alfa diminished viral replication.

These results suggest the role of IRE1alpha/XBP1 in the replication of ICOV and in favoring *Leishmania* growth during coinfection.

### 3.4. ICOV/Leishmania Coinfection Increases the Expression of XBP1-Regulated Genes Involved in the Synthesis of Phosphatidylcholine

The expansion of the ER demands phospholipids, such as phosphocholine, which is abundant in mammalian cell membranes [35]. Since XBP1 regulates a set of genes involved in the phospholipid synthesis, we decided to address whether the ER stress-induced during the infection by ICOV and *L. amazonensis* would upregulate the expression of the cholinephosphotransferases CPT1 and CEPT1enzymes, which regulate the final steps of the Kennedy pathways [36]. Figure 4A,B shows the augmented mRNA levels of CHPT1 during ICOV infection and upon the coinfection, respectively. Notably, the coinfection resulted in a slight but significant increase in the expression compared to the single infection by *L. amazonensis* or ICOV. In Figure 4C,D, we observed a similar mRNA expression profile for CEPT1. Significantly, the chemical inhibition of IRE1α abolished the expression of CHP1 and CEPT1. The enhanced expression of CHPT1 and CEPT1 is sustained for 24 h after coinfection (Appendix A).

We concluded that XBP1 is transcriptionally active in our models and may regulate the expression of genes involved in ER membrane expansion.

### 3.5. ICOV and the Coinfection with L. amazonensis Engage the ER Stress Branch PERK/ATF4

Then, we addressed if ICOV and the coinfection with *L. amazonensis* would increase the mRNA levels of ATF4 and NRF2, regulated by the ER stress sensor PERK. Figure 5A,B, respectively, shows the ICOV upregulation of ATF4 and enhanced levels during the coinfection. NRF2 is a master transcription factor that regulates the expression of genes involved in the antioxidative stress response [37], induced via the PERK/ATF4 pathway. Figure 5C,D shows the upregulation of NRF2 by ICOV expression due to the coinfection. The pharmacological inhibition of PERK diminished the mRNA levels of ATF4 and NRF2, thus demonstrating the dependence of PERK activation. We observed a significant increase in ATF4 and NRF2 mRNA levels during the coinfection, which is sustained for 24 h, suggesting the potentiation of the antioxidative response in this condition. (Appendix A).

## 4. Discussion

Phleboviruses comprise a group of three segmented RNA viruses transmitted to vertebrates by Phlebotomine sandflies and other arthropods, such as ticks [38]. Approximately 67 phleboviruses species are recognized, and they share a high degree of amino acid identity in the RNA-dependent RNA polymerase coded in the L RNA segment genome [23].

Phlebotominae species are the primary vectors of *Leishmania*, which are obligatory intracellular parasites in vertebrates, including humans. The human infection by *Leishmania* species may cause a plethora of symptoms, varying from limited self-healing cutaneous ulcers and deforming mucosal lesions to lethal visceral conditions [39]. Transmission of cutaneous Leishmaniasis and phleboviruses overlap geographically [40], and molecular PCR tests revealed the coinfection of a limited set of *Phlebotomus* in Europe [41]. A growing body of in vitro and in vivo evidence suggests the potential importance of Phleboviruses in the aggravation of Leishmaniasis and the enhancement of parasite replication [30,42,43].

Our TEM studies revealed some aspects of the interaction of ICOV with macrophages, which are the primary cell type infected by *Leishmania* parasites. We could not attest that the viral particles shown in Figure 1C,D were within a large vacuole since the image refers to a 60 µm slice of the sample. In the same way, the belt-like structure observed in Figure 1E suggests a disconnection in progress between both. However, if what we noticed as a vacuole was, indeed, part of the macrophage plasma membrane projection, the phenomenon could result from the initial mechanisms that lead to virus internalization. Future tomography analysis of similar virus-cell interactions may resolve these uncertainties. The presence of Phleboviruses in canyons at the base of cell membrane projections, in sites where the plasma membrane is discontinued (Figure 1F), is suggestive of viral egress.

ICOV was isolated from the rodent *Nectomys squamipes*, which may serve as an *L. (L.) amazonensis* reservoir, and it shares the same environment as the vector *Lutzomyia flaviscutellata*, the primary vector of *L. amazonensis*. Some reports suggest that ICOV may occur in a broad geographical distribution in Brazil, and serological studies revealed the occurrence of ICOV in distinct vertebrates, such as ruminants and non-human primates [44].

Viruses may induce the ER stress response to adapt the host cell machinery to produce new synthesized viral particles [13,32]. Under viral infection, some branches of ER stress are activated, reprogramming the transcriptional and translational programs [32]. The expression of genes involved in synthesizing the phospholipids under XBP-1 control and the transcription of antioxidative response genes mediated by the activation of NRF2 is a common feature of the induced ER stress program.

To fully ascertain that ER stress is occurring in our models and the majority of the literature, measuring the mass of unfolded or misfolded proteins inside the ER would be necessary. We did not address this point. However, we could establish the engagement of ER stress sensors, PERK and IRE1, in addition to the processing of XBP1 and the expression of some genes regulated in response to the ER stress response. Our data support the notion that ICOV and Leishmania infection cooperatively mobilize the ER stress program. We demonstrated that ICOV induces the XBP-1 and PERK/ATF4 mediated arms of ER stress, and the pharmacological inhibition of IRE1 or PERK reduces the viral replication in macrophages. The coinfection with *L.amazonensis* potentiated and sustained the ER stress response, apparently leading to a mutual benefit in viral and parasite *L. amazonensis* infections may be limited in humans, causing localized cutaneous lesions. More rarely, this species is found in patients with the anergic diffuse cutaneous leishmaniasis (ADCL), with a poor TH1 response and abundant amastigote forms [45]. Previously, we showed increased levels of the dsRNA sensor PKR and IFNb in the lesions of ADCL patients [46,47]. It is attempting to hypothesize that viral RNA coinfection may be implicated in the outcome of the infection. The coinfection in vivo assays demonstrated with ICOV (28) revealed high levels of IFNβ and IL-10, which may partially mirror the finding in some clinical studies [45]. Our findings suggest that the ER stress response is one component that may shape the parasite’s replication inside macrophages. Currently, we are addressing the coinfection in cutaneous leishmaniasis and the expression of ER stress markers’ survival and growth inside the host cell. This finding might be of epidemiological importance considering that wild vertebrate reservoirs may host both pathogens and mutually cooperate to increase the supply of infective forms to sandflies and maintain the natural transmission cycle between vectors and vertebrates.

In conclusion, our findings corroborate the notion that the ER stress response embases the cellular processes involved in *Leishmania* and ICOV intracellular replication, and it enlights the fundamental molecular mechanisms that can govern the coinfection between phleboviruses and *Leishmania*.

## Figures and Tables

**Figure 1 viruses-14-01948-f001:**
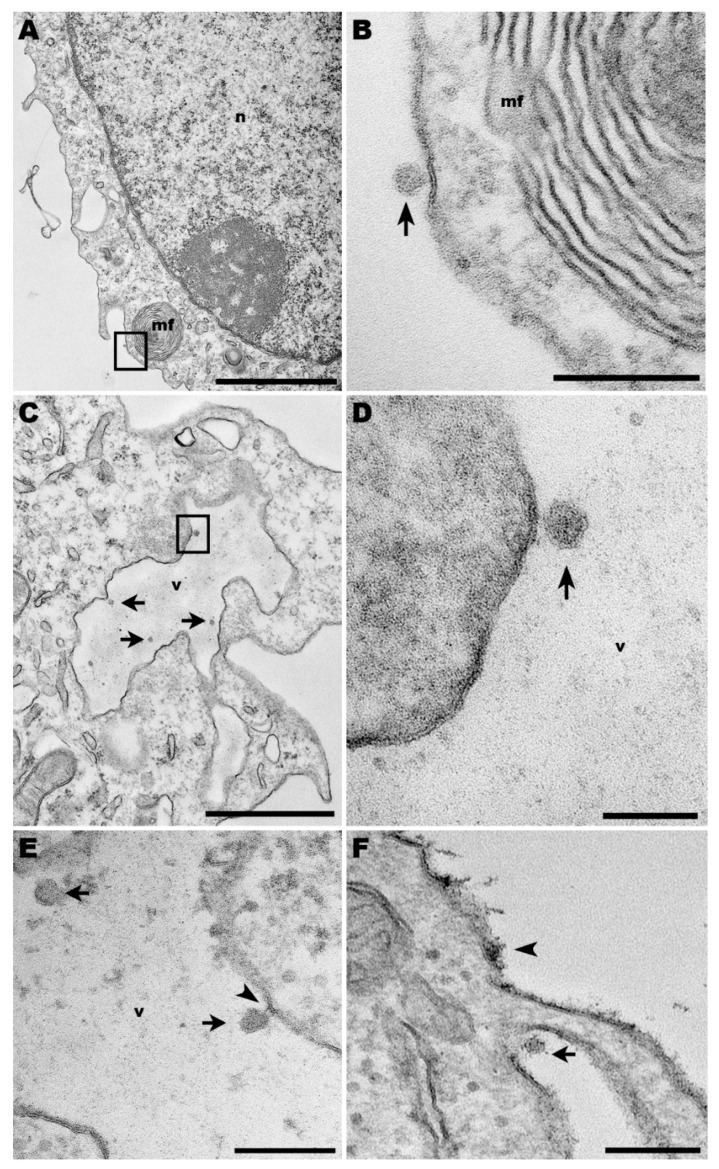
Transmission electron microscopy of macrophages infected with Phlebovirus ICOV. (**A**,**B**) ICOV particle on the surface of an infected macrophage from C57BL/6 mice (arrow), near of a myelin figure (mf) at the periphery of the cell (inset). (**C**,**D**) Inside virus-infected macrophages, ICOV (arrows and inset) were observed within large vacuoles (v). (n) nucleus. (**E**) Virus-containing vacuoles (v) displayed ICOV in the lumen (arrows) or linked to the vacuole membrane, through a membrane belt (arrowhead). (**F**) Viruses (arrow) were also observed in canyons of cell membrane projections. Bars: (**A**) 2 µm; (**B**,**E**,**F**) 200 nm; (**C**) 1 µm; (**D**) 100 nm.

**Figure 2 viruses-14-01948-f002:**
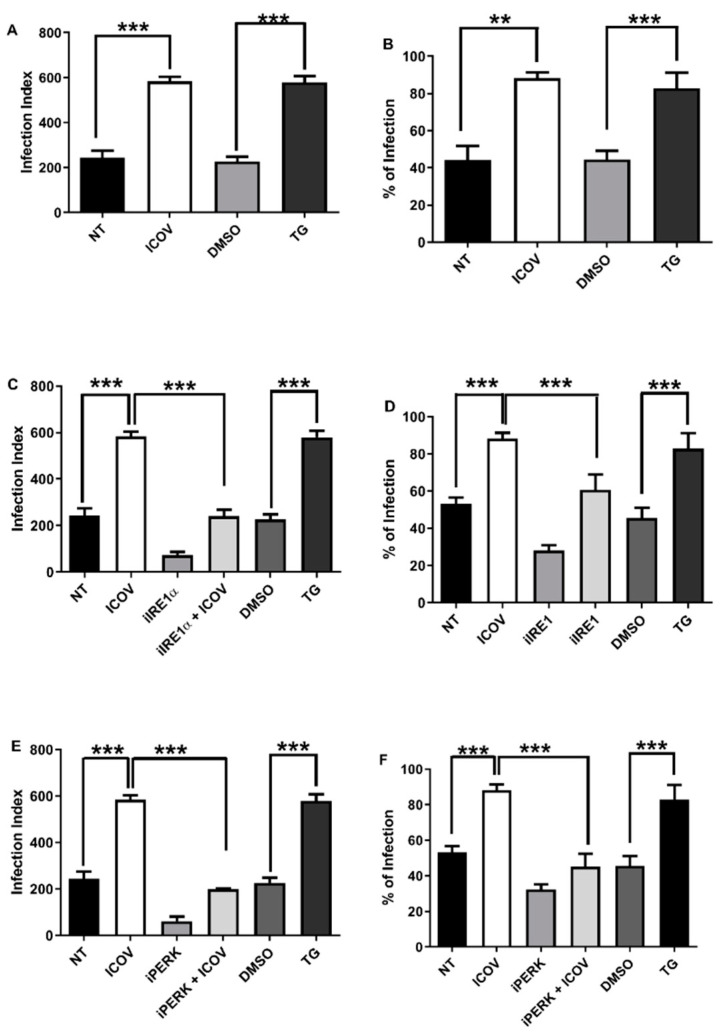
The replication of *L. amazonensis* is favored during ICOV coinfection in an ER stress pathway dependent manner. Peritoneal macrophages from C57BL/6 mice were not treated (NT), infected with *Phlebovirus* Icoaraci (ICOV) at MOI 1, treated with 1 µM Tapsigargin (TG) for 1 h (**A**,**B**) or DMSO and pre-treated with 100 nM of IRE1a inhibitor during 1 h (**C**,**D**) or with 40 nM of PERK inhibitor during 1 h (**E**,**F**) followed by ICOV infection. After, all cells were infected with stationary-phase promastigotes of *L. amazonensis* at a ratio of 5 parasites/cell. At 48 h pi, one hundred Giemsa-stained cells were inspected, and the infection index was calculated (percentage of infected macrophages multiplied by the average number of amastigotes per macrophage) and the percentages of infected macrophages as indicated in the figure. Asterisks indicate significant differences between groups by ANOVA, with ** *p* < 0.001 or *** *p* < 0.0001.

**Figure 3 viruses-14-01948-f003:**
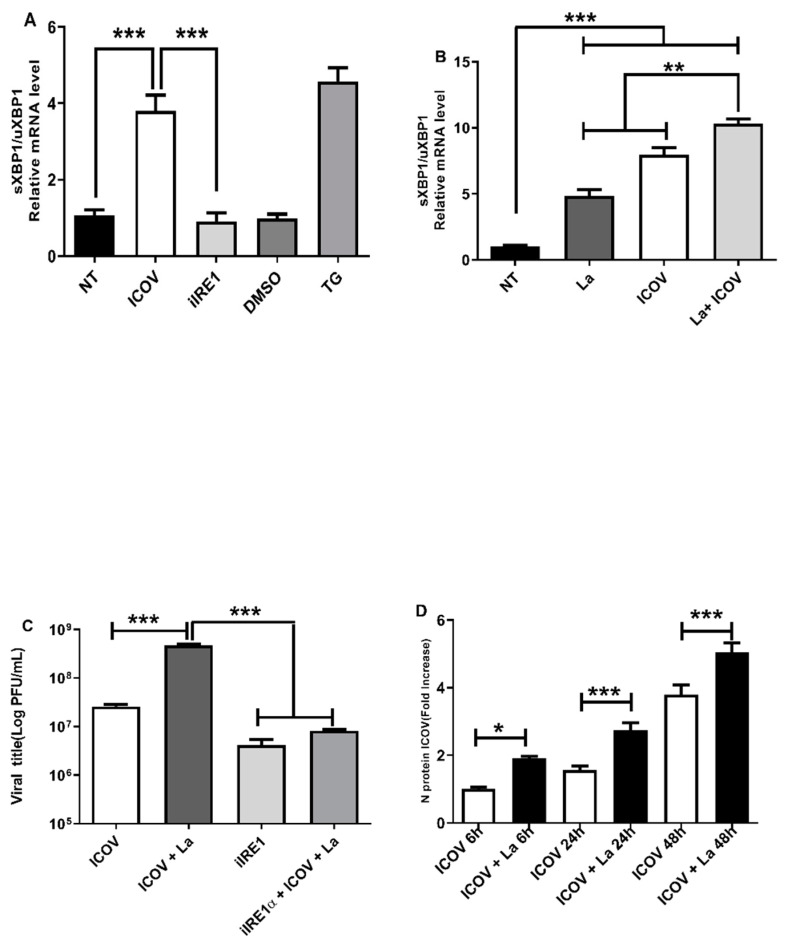
ICOV replication depends on ER stress via and is favored during *L. amazonensis* coinfection. Peritoneal macrophages from C57BL/6 mice were not treated (NT), infected with Phlebovirus Icoaraci (ICOV) at MOI 1, pre-treated with 100 nM of IRE1α inhibitor during 1 h followed by ICOV infection or treated with 1 µM Tapsigargin (TG) or DMSO (**A**). After 6 h, sXBP1 transcript levels were measured by qRT-PCR normalized to uXBP1 transcript levels. The same analysis was done during ICOV/*L. amazonensis* coinfection (**B**). The viral title was analyzed through plaque assay (**C**). The mRNA levels of ICOV N protein was verified by qRT-PCR in peritoneal macrophages coinfected with ICOV and *L. amazonensis* as indicated in the figure (**D**). Asterisks indicate significant differences with * *p* < 0.01, ** *p* < 0.001 or *** *p* < 0.0001.

**Figure 4 viruses-14-01948-f004:**
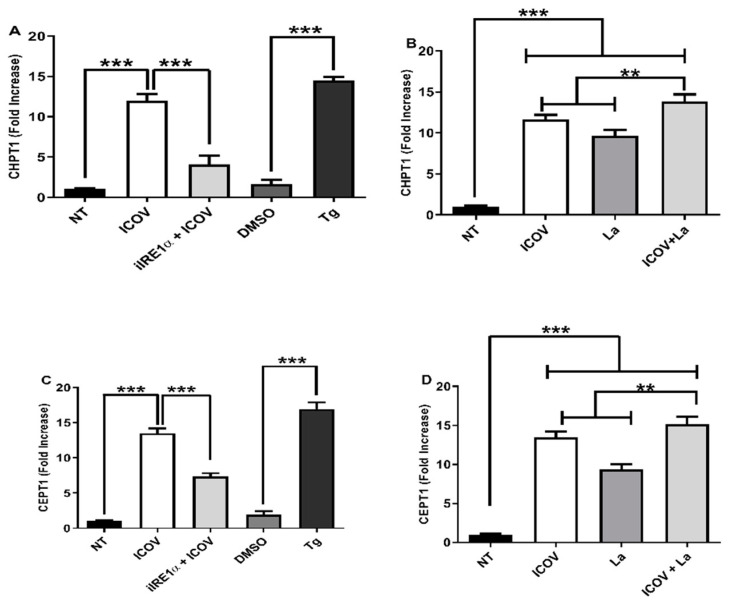
Upregulation of XBP1 dependent genes during ICOV and ICOV/*L. amazonensis* coinfection. CHPT1 (**A**,**B**) or CEPT1 (**C**,**D**) transcript levels were measured by qRT-PCR normalized to GAPDH transcript levels after 6 h of treatment and/or infections. Peritoneal macrophages from C57BL/6 mice were not treated (NT), infected with *Phlebovirus* Icoaraci (ICOV) at MOI 1, pre-treated with 100 nM of IRE1a inhibitor during 1 h, followed by ICOV infection or treated with 1 µM Tapsigargin (TG) or DMSO (**A**,**C**). The same analysis was done during ICOV/*L. amazonensis* coinfection (**B**,**D**). Asterisks indicate significant differences with ** *p* < 0.001 or *** *p* < 0.0001.

**Figure 5 viruses-14-01948-f005:**
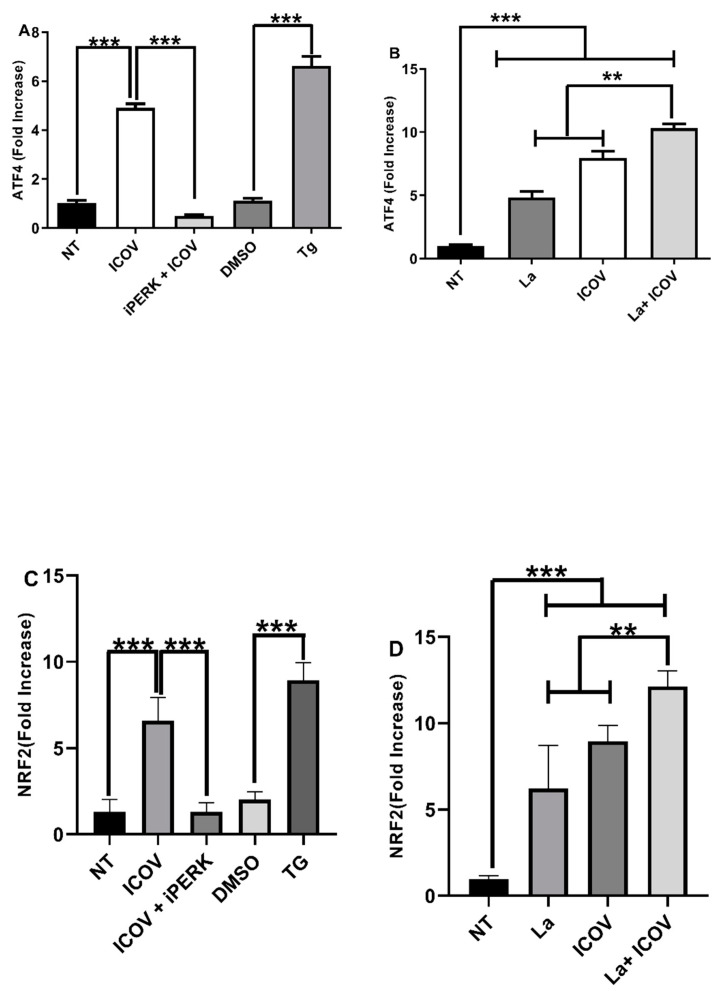
PERK/ATF4 pathway is required during ICOV and the coinfection with *L.* amazonensis. ATF4 (**A**,**B**) or NRF2 (**C**,**D**) transcript levels were measured by qRT-PCR normalized to GAPDH transcript levels after 6 h of treatment and/or infections. Peritoneal macrophages from C57BL/6 mice were not treated (NT), infected with Phlebovirus Icoaraci (ICOV) at MOI 1, pre-treated with 40 nM of PERK inhibitor during 1 h, followed by ICOV infection or treated with 1 µM Tapsigargin (TG) or DMSO (**A**,**C**). The same analysis was done during ICOV/*L. amazonensis* coinfection (**B**,**D**). Asterisks indicate significant differences with ** *p* < 0.001 or *** *p* < 0.0001.

## Data Availability

All relevant data are within the manuscript and its Appendix A.

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
