# Peer review of "Endoplasmic Stress Affects the Coinfection of Leishmania Amazonensis and the Phlebovirus (Bunyaviridae) Icoaraci"

_viruses, 2022, doi:10.3390/v14091948_

Round 1

Reviewer 1 Report

I have reviewed the paper entitled “ENDOPLASMIC STRESS AFFECTS THE COINFECTION OF LEISHMANIA AMAZONENSIS AND THE PHLEBOVIRUS (BUNYAVIRIDAE) ICOARACI” and found the information given here interesting. I have however some comments. Authors are mainly talking about sand flies/Leishmania, but there is no epidemiological introduction about it. I suggested authors adding leishmania life cycle, related symptoms from patients, current aspects on epidemiology and treatment into introduction section. May the obtained results provide any control strategies or provide any recommendations for endemic areas where the coinfection exists? Why authors select the used Leishmania and virus species in the present study? I think authors should cite the limits of the laboratory work since environmental factors have been reported to affect parasite development and insects’ physiological state and immune response. Any data about the endoplasmic stress and coinfection with symbiotic microorganisms since virus is basically infectious agents for vectors rather than symbiotic microorganisms although they can affect vector competence?

Reviewer 2 Report

The aim of this study was to test how viral coinfection with Phlebovirus Icoaraci, (an inductor of the endoplasmic reticulum stress program in macrophages) overlaps and potentiates L. amazonensis infection, potentiating the endoplasmic reticulum stress and thus modulating the severity of this parasitic disease, the human cutaneous leishmaniasis.

All in all, the manuscript herein presented can be interesting as a proof of concept of the validity of the above-mentioned aim. However, some issues should be discussed in deepness to gain more information from the presented data.

  1. The first issue that should be clarified is the fact that the authors claim to be measuring endoplasmic reticulum stress. However, from what is described, what is being measured herein are endoplasmic reticulum stress markers such as the different genes that are being modulated by the herein described procedures. So please clarify and expand this important discussion.
  2. On the other hand, the authors describe that they use thioglycolate to stimulate and elicit the macrophages to be used. Regarding this methodology it is important to clarify if the thioglycolate used is Brewer or NIH. Also, a brief discussion on why resident macrophages were not used seem to be important. The results can be very different for each case, as has been described recently by Pavlou, S. et al. (J Inflamm 14, 4 (2017). https://doi.org/10.1186/s12950-017-0151-x)
  3. Besides that, how coinfection with Leishmania and the virus was performed is not herein thoroughly described. That description is fundamental to better understand the results. Only in the figure legend of Fig 3 a brief description appears.
  4. Additionally, is there a reference to mention in relationship with Phlebovirus production, quantification and infection. If so, please include it. Please also explain how a MOI of 0.01 or 1 was decided to be used in each case (production of viruses and infection of macrophages).
  5. The authors use a numbering that is confusing. Fig 1G does not exist. Please correct.
  6. In Fig 1D the authors explain that “ICOV seemed to be also present in the lumen of large vacuoles”. However, in the discussion they doubt about this result. Please clarify, especially since no markers for vacuole were used.
  7. Only in the figure legend of Fig 2 the authors identify the origin of the macrophages (C57BL/6 mice). This should be stated in the methods. Also, a brief discussion or even better, results would be wonderful to have, with macrophages coming from BALB/c mice, due to the so completely different behavior and immunological responses elicited by these two types of mice.
  8. Finally, the authors conclude that endoplasmic reticulum stress responses are involved in the cellular processes involved in Leishmania and ICOV intracellular replication and identifies which markers are modulated in response to the challenge. However, a deeper discussion of the real outcome of these results may add value to the presented data. Especially in the context of the type of human leishmaniasis produced by L. amazonensis and the severity of the disease that could be reached.

Round 2

Reviewer 2 Report

I am here by sending the second  review of the manuscript "ENDOPLASMIC STRESS AFFECTS THE COINFECTION OF LEISHMANIA AMAZONENSIS AND THE PHLEBOVIRUS (BUNYAVIRIDAE) ICOARACI" by: José V. dos Santos, Patricia F. Freixo, Áislan de C. Vivarini, Jorge M. Medina, Lucio A. Caldas, Marcia At-5 tias, Karina L. Dias Teixeira, Teresa C. C. Silva, Ulisses G. Lopes. This manuscript was sent to the Journal Viruses for its evaluation and publication.

 As previously mentioned, the aim of this study was to test how viral coinfection with Phlebovirus Icoaraci, (an inductor of the endoplasmic reticulum stress program in macrophages) overlaps and potentiates L. amazonensis infection, potentiating the endoplasmic reticulum stress and thus modulating the severity of this parasitic disease, the human cutaneous leishmaniasis.

The manuscript herein presented is interesting as a proof of concept of the validity of the above-mentioned aim. The authors have answered appropriately the queries raised and I congratulate them for the manuscript. It opens new questions regarding co-infection and stress response.